# Position: The Alignment Community is Unintentionally Building a Censor's Toolkit

**Sarah Ball** [* 1 2]  **Phil Hackemann** [*]

## Abstract

This position paper argues that modern AI alignment methods – originally designed to prevent harmful output – are dual-use technologies that may easily be misused by malicious actors for censorship and manipulation. By mapping current alignment techniques to the possibility and actual cases of misuse, we show that the quest for a "perfectly aligned" model inadvertently also provides malicious actors with an ever-improving tool for informational dominance. We need to discuss this dual-use potential *now*, as its risk is exacerbated by rapid user adoption of AI as information provider, economic power asymmetries, and a political landscape that increasingly shifts towards authoritarianism. We conclude by urging the community to consider the intentional misuse of AI alignment mechanisms and propose mitigation strategies to safeguard against this dual-use potential.

## 1. Introduction

We in the alignment community have always been confident that we are on the right side of history: we are the ones building the safeguards that will protect humanity from the risks of AI. This mission has driven the development of increasingly sophisticated techniques to robustly align models with human values, fueled in part by an ongoing arms race against jailbreaks, prompt injections, and fine-tuning attacks (Chao et al., 2024; Debenedetti et al., 2024; Qi et al., 2024; Yi et al., 2024). Over time, our field has learned to recognize the *unintended* harms our methods can cause – how alignment data that fails to represent diverse populations leads to the systematic marginalization of underrepresented groups (Kirk et al., 2024a;b; Wu et al., 2025). This recognition

has prompted valuable work on pluralistic alignment (Ryan et al., 2024; Santurkar et al., 2023; Sorensen et al., 2024) and technical innovations to better capture the full spectrum of human values (Chakraborty et al., 2024; Mukherjee et al., 2025; Sun et al., 2025).

Yet, what has received far less attention are the *intended* negative consequences when our alignment methods are placed in the wrong hands. The history of science teaches us that well-intentioned tools can be misused, such as research in nuclear physics gave us both energy *and* the atomic bomb. Similarly, we must now confront an uncomfortable truth: **The alignment methods we develop are dual-use technologies, and they are already being weaponized for censorship and manipulation**.

**Why Now?** Growing numbers of users turn to AI for information retrieval (Simon et al., 2025; Tamkin et al., 2024), granting these models unprecedented power to shape public knowledge and opinion. This shift coincides with a political landscape moving towards more authoritarian regimes (Nord et al., 2025) and an AI economy increasingly dominated by a few powerful actors, exacerbating the misuse risk of AI alignment methods (Vesteinsson et al., 2025).

**How is This New?** Prior work on AI risks addresses *general AI systems* rather than *technical alignment methods*, typically from a high-level and often hypothetical angle (Berryhill et al., 2024; Hendrycks et al., 2023; Saeri et al., 2024). While some work does move beyond the hypothetical, documenting isolated instances of censorship in specific models or regions (Huang et al., 2025; McDonell, 2023; Naseh et al., 2025; Noels et al., 2025; Qiu et al., 2026) none systematically frames these as instances of a broader dual-use risk inherent to alignment methods themselves. In contrast, we *i)* systematically map alignment techniques to their dual-use potential, showing *how* and by *whom* each method can be repurposed for informational control, *ii)* present evidence that this weaponization is already ongoing, *iii)* analyze the current social, economic, and political ecosystem that makes such misuse increasingly likely and consequential, and *iv)* develop suggestions of how to face these risks, calling for competitive model pluralism, improved auditing for verifiable alignment, public education, and genuine researcher reflection on dual-use implications of their work.

---

[*]Equal contribution  [1]Department of Statistics, LMU Munich, Munich, Germany  [2]Munich Center for Machine Learning (MCML), Munich, Germany. Correspondence to: Sarah Ball <sarah.ball@stat.uni-muenchen.de>.

*Proceedings of the 43rd International Conference on Machine Learning*, Seoul, South Korea. PMLR 306, 2026. Copyright 2026 by the author(s).

**In a Nutshell.** We do not argue for halting alignment research – it is necessary to prevent serious harm. Rather, we contend that given current societal, economic, and political developments worldwide, the alignment community must actively consider how our methods can be weaponized, not just perfected. The methods we refine today will determine how information is controlled tomorrow.

## 2. The Dual-Use Problem of AI Alignment

As a scientific community, we have grown accustomed to unquestioningly interpreting the term "*alignment*" in an inherently positive way. Yet, this interpretation obscures a critical fact: the technical methods of alignment are in fact purpose-agnostic tools that can be used for various goals. This becomes evident in the definition offered by Ji et al. (2025): *"AI alignment aims to make AI systems behave in line with human intentions and values."* Certainly, these "human intentions and values" to which a model is aligned are *often* those that are generally recognized as 'good', like preventing "the spread of hate speech, misinformation, and dangerous instructions" (Noels et al., 2025). But there is nothing inherent in alignment methodology that guarantees benevolent outcomes. Whoever defines those "intentions and values" fundamentally determines whether the resulting system serves safety or oppression (Winner, 1980; Zhi-Xuan et al., 2025).

**What if a Malicious Actor Defines Those Values?** Accordingly, if in the hands of a malicious actor, a well-intentioned system can easily be transformed into a machine for censorship and manipulation using the very same alignment mechanisms. Hence, just as other scientific advances and new technologies have shown dual-use potential (we already mentioned nuclear energy as an example), also we in the alignment community must confront the possibility that the ever-improved safeguards we engineer may unintentionally become instruments of information control for others.

### 2.1. What are the Threats We are Talking About?

To understand how alignment methods can be weaponized, we must examine the two primary vectors through which a dominant actor can exert control over a model's output: the total suppression of information (censorship) and the distortion of user perception (manipulation).

**Censorship.** Censorship is "a system in which an authority limits the ideas that people are allowed to express and prevents [...] communication[s] from being seen or made available to the public, because they include or support certain ideas" (Cambridge Dictionary, a). In AI systems, censorship renders specific information inaccessible to users (Sheehan, 2023). Alignment methods are currently used to, for example, prevent LLMs from releasing bomb-making instructions. However, in the same technical way, they can also be made to withhold information about historical facts or political opinions.

**Manipulation.** Another, more subtle danger than censorship is manipulation, meaning "influencing or controlling someone or something to your advantage, often without anyone knowing it" (Cambridge Dictionary, b). Accordingly, LLMs could be steered to distort facts or give biased answers. Systematically applied via alignment, this can lead to the mass manipulation of opinions and preferences.

### 2.2. What is Misuse?

Given that every form of alignment suppresses or alters the output in some way, the question arises as to which instances actually constitute "misuse". Not all use cases falling under the definitions of censorship and manipulation above are unambiguous; for instance, different societies hold varying views on where legitimate free speech ends and punishable hate speech begins. In these cases, the assessment – including the scientific one – will always be influenced by one's own sociocultural background. Nevertheless, we consider certain applications to qualify as misuse even under the strictest definition of the term: namely, when they (i) violate internationally recognized human rights, including freedoms of expression, thought, and access to information as enshrined in the Universal Declaration of Human Rights (1948), to which virtually all states are formally committed, or (ii) serve the private interests of a small number of powerful actors at the expense of the billions of users who have no visibility into, or recourse against, those decisions. These are the concrete threats we want to raise awareness of.

### 2.3. Who are Possible Malicious Actors?

To discuss the dual-use risk of alignment tools, we must identify who holds the 'keys' to these instruments. Two entities are currently positioned to dictate AI behavior at scale: state actors and foundation model providers. These groups are distinguished by their specific levers of power, available resources, model access, and reach.

**State Actors.** Governments possess unique coercive power to mandate alignment objectives through legislation and enforcement. State actors command substantial computational resources and can recruit top technical talent. While they typically lack direct model development capabilities, they maintain indirect control through regulatory frameworks that can mandate exclusive deployment of approved models or prohibit alternatives. Their influence operates primarily through intermediaries (model providers), but carries the force of law. Emerging AI legislation in many jurisdictions already contain binding rules for AI alignment. These include compliance with applicable laws, as well as prevention of harmful content and breach of privacy (Naseh

et al., 2025). In the same way, however, it is also possible to use such a legal framework to extend the scope of 'harmful content' to further use cases, possibly under the pretext of 'national security'. Accordingly, authoritarian regimes could use this lever for compelling model providers to align their systems with state-approved viewpoints, potentially suppressing dissent or marginalizing minority groups.

**Foundation Model Providers.** Organizations that train foundation models – typically large tech companies – exercise the most direct control over alignment processes. They make fundamental decisions regarding training data composition, pre-training objectives, and alignment methodologies. By maintaining elite technical teams and substantial computational infrastructure, these providers enable sophisticated alignment interventions that reach millions of users worldwide. While third-party actors (downstream fine-tuners) can also adapt these models for specific ideological or domain-specific objectives, our arguments and analyses focus primarily on the original model developers. Unlike fine-tuners, who face significant resource constraints and limited reach, these companies wield extraordinary, and often unchecked, power. Although constrained by regulatory frameworks, the centralized nature of their control presents a unique risk for potential misuse (Arogyaswamy, 2020).

### 2.4. How Realistic are These Threats?

Authoritarian regimes have a high interest in and are known for tightly controlling information systems and especially the internet, including online newspapers, social media, and search engines, in order to maintain control over political narratives (Dong, 2012; Islas-Carmona et al., 2024; Kravets & Toepfl, 2022; Xu & Albert, 2017). Given the emergent role of AI as an increasingly important source of information and crucial determinant for public perception (see discussion in Section 4.1), this will certainly also apply to LLMs.

Attempts to politically influence the direction of AI models can already be observed worldwide. For example, China has implemented strict laws to control AI development, use, and output (Lin, 2024; Noels et al., 2025; Sheehan, 2023). Namely, the *Provisional Administrative Measures of Generative Artificial Intelligence Services*, enacted by the Cyberspace Administration of China (CAC) in August 2023, mandates that AI-generated content must align with "socialist core values" and prohibits content that threatens national unity, social stability, or government authority, as well as output "inciting subversion of national sovereignty or the overturn of the socialist system", and "harming the nation's image" (CAC, 2023; McMorrow & Hu, 2024). How this practically materializes can be witnessed in Chinese chatbots like *DeepSeek* and Baidu's *Ernie Bot*, which are systematically made to refuse discussing politically sensitive topics, like the 1989 Tiananmen Square massacre, while

amplifying Chinese Communist Party positions, such as on Taiwan and U.S. relations (Huang et al., 2025; Li, 2025; McDonell, 2023; Naseh et al., 2025; Qiu et al., 2026).

Similar examples of state intervention and attempts to develop own LLMs that are aligned with the government's political agenda have been reported in Vietnam, Thailand, Russia, Belarus, and Iran, among others (Vesteinsson et al., 2025). But also in democracies like the U.S., we can observe increasing pressure from the Trump administration on tech companies to modify their models in line with his agenda; for example, by removing instructions regarding diversity, equity, and inclusion (Robins-Early & Gambino, 2025; Roose, 2025).

The risk of political manipulation is not exclusive to state actors, though: also model providers may unilaterally instrumentalize alignment mechanisms to enforce personal ideological agendas or interests. For instance, Elon Musk publicly announced to "fix" specific outputs from his own LLM, *Grok*, with which he disagreed (e.g., regarding natalism, left-wing violence, or non-binary genders). Following these interventions, the model's tone shifted quickly. Moreover, it was purposely realigned to promote his specific political agenda – such as claims regarding an alleged white genocide in South Africa (Conger, 2025; Robins-Early & Gambino, 2025; Thompson et al., 2025).

The stakes are therefore substantial. We should not underestimate the impact misused or misguided alignment efforts can have. Alignment has the potential to (quietly) shift models from informational tools to normative gatekeepers that influence entire societies.

## 3. How Alignment Techniques Can Be Misused

To substantiate our argument, we next examine the most commonly applied alignment methods for their dual-use potential. Modern frontier LLMs are typically shaped by a three-stage control stack: i) *pre-training data curation* (what enters the base model), ii) *post-training alignment* (how the model is tuned to follow instructions and preferences), and iii) *inference-time interventions* (run-time policies and additional guardrails). In the following, we describe these control elements and analyze their dual-use potential with respect to requirements for access, computational resources, technical expertise, as well as the ease and depth of modification (see Table 1 for an overview). We further underpin our analyses with empirical evidence of current weaponization.

### 3.1. Pre-Training Data Filtering

Pre-training instantiates a model's world knowledge through exposure to large-scale data corpora. This pre-alignment corpus gets filtered to improve data quality and reduce unwanted content. Established filtering practices comprise

*Table 1.* Characteristics of alignment methods and their dual-use potential.

| Dual-use characteristics | Pre-training filtering | Post-training alignment | Inference-time control |
|---|---|---|---|
| Access requirements | pre-training pipeline | model weights | runtime access |
| Computational resources | very high | moderate–high | negligible–moderate |
| Technical expertise | high | moderate–high | low–moderate |
| Ease of modification | moderate–difficult | moderate | easy |
| Depth of modification | fundamental | persistent | superficial |

de-duplication and the removal of personally identifiable information, as well as domains with unsafe, adult, low-quality, and boilerplate content (Grattafiori et al., 2024). This typically happens through a combination of *heuristic filtering* and *model-based filtering* using trained classifiers to detect high-quality content. One heuristic filtering method is, for instance, "dirty word" counting to filter out adult websites not mentioned on domain block lists (Raffel et al., 2020).

**Dual-Use Potential.** Data filtering methods enable the selective removal of content that is deemed unwanted, enabling the targeted suppression of information. Yet, when assessing the ease of misuse, significantly modifying pre-training data and re-training the model from scratch is feasible only for well-resourced actors (both with respect to compute and expertise) with access to the complete training infrastructure. While heuristic-based filtering (e.g., keyword matching, domain blocking) can more easily remove specific facts or historical events, model-based filtering, which can excise more abstract concepts or viewpoints, requires greater technical sophistication but is increasingly accessible through improved language models (Chen et al., 2025). Although being the most effortful, pre-training interventions produce fundamental changes to model knowledge: information absent from pre-training cannot be generated without explicit post-training instruction or additionally provided context.

**Real-World Evidence.** A Financial Times article in summer 2024 explains how Chinese AI engineers have become increasingly sophisticated in censoring LLMs: As a first step in the pipeline, Chinese AI companies filter out "problematic" information and build a dataset of keywords, similar to the heuristics-based filtering methods described before. These problematic keywords violate "core socialist values" like "inciting the subversion of state power" (McMorrow & Hu, 2024). To further support these efforts, the government has been building its own training datasets. In 2023, Hugging Face was blocked by Chinese authorities, while datasets like the "mainstream values corpus" developed by the People's Daily, the Communist Party's official newspaper, were being created to reflect what party leaders consider safe content for training local AI models (Lin, 2024). In addition to these internal engineering efforts, this induces

an even broader systemic effect: by filtering or manipulating publicly available data, one can *indirectly* influence *other* models that use this data for training, without their developer's knowing. This is why even Western LLMs were shown to apply self-censorship when prompted in Simplified Chinese, because the available training corpus is heavily influenced by the Chinese government (Ahmed & Knockel, 2024; Noels et al., 2025; Waight et al., 2026). Hence, while these models were not necessarily intended to censor, their behavior (in a particular language) reflects a training foundation that has been fundamentally compromised.

### 3.2. Post-Training Preference Alignment

Post-training usually aims to turn a base pre-trained model into a "helpful, honest and harmless" assistant (Askell et al., 2021). A predominant framework for achieving this is *Reinforcement Learning from Human Feedback* (RLHF; Bai et al., 2022a; Christiano et al., 2017; Ouyang et al., 2022). RLHF involves the collection of human preferences from a selected pool of annotators or users, followed by training a reward model to predict these preferences. The reward model is then used to optimize the policy model using Proximal Policy Optimization (Schulman et al., 2017), or variants like Group Relative Policy Optimization (Shao et al., 2024). Another common framework is what we call *guideline-based alignment*. Variants within this framework integrate explicit guidelines into the process, removing the need for human feedback collection. In "Constitutional AI" (Bai et al., 2022b) a set of principles guides critique-and-revision, and the resulting preferences are used for the reinforcement learning step. Similarly, one of the newest alignment paradigms, "Deliberative Alignment" (Guan et al., 2024), uses a pre-defined set of policies that the model is taught to reason over before answering. This approach does not need human-written chains-of-thoughts or preferences for safety alignment.

**Dual-Use Potential.** RLHF and related techniques align models to specified preferences, but these same methods can also enable alignment to single actors' interests, possibly resulting in censorship and manipulation. Post-training requires access to models and moderate computational resources – substantially less than pre-training but more than

inference-time methods. Fine-tuning is accessible to model providers and potentially to downstream actors with sufficient compute. They exercise complete control over preference data collection, annotator pool selection, and reward model training – all levers to steer alignment in specific directions. Concretely, by curating preference datasets to favor one's own ideology, and by carefully briefing and selecting only well-disposed annotators, it is possible to effectively steer a model's behavior to promote one's own interests. Guideline-based approaches (like Anthropic's Constitutional AI and OpenAI's Deliberative Alignment) offer even greater flexibility: providers can modify ethical guidelines with less technical overhead, making iterative alignment changes easier compared to approaches requiring preference data curation and reward model retraining. Post-training methods directly modify model parameters; the changes are, however, more shallow than pre-training modifications and can be trained away or circumvented using methods from the adversarial attacks literature (Lee et al., 2024; Qi et al., 2024). Nevertheless, these interventions can effectively enforce specific viewpoints while systematically refusing to engage with other perspectives.

**Real-World Evidence.** China's cyberspace regulator CAC reportedly requires model providers to prepare between 20,000 and 70,000 questions to test whether models produce "safe" responses, alongside a dedicated refusal dataset of 5,000 to 10,000 prompts that models must decline to answer (Lin, 2024). Notably, approximately half of these refusal targets focus on political ideology and criticism of the Communist Party. Furthermore, in May 2024, CAC revealed plans about a chatbot that adheres to political guidelines using, among other things, the 14-point political philosophy of Chinese leader Xi (Lin, 2024). These developments indicate that governments utilize *guidelines* or *refusal datasets* to steer model alignment toward state-approved narratives, effectively repurposing post-training alignment for ideological compliance.

### 3.3. Inference-Time Control

Even after post-training, deployed systems typically apply inference-time controls that can substantially affect behavior. To this end, chat-style deployments often prepend hidden *system prompts* with instructions that define role, priorities, and constraints. This can be understood as a high-leverage, deployment-time alignment layer. In addition, model providers commonly use separate *safety classifiers* (before and/or after generation) to detect and stop disallowed content (Inan et al., 2023; Microsoft Corporation, 2025; NVIDIA Corporation, 2025).

**Dual-Use Potential.** Inference-time interventions modify or stop model outputs without altering underlying parameters, offering the lowest barrier to deployment but also the most

superficial control. System prompts are easily accessible for model providers and require negligible computational cost, making them the most accessible intervention point. They can be modified instantaneously without specialized expertise, enabling rapid iteration on alignment objectives. Classifiers require training infrastructure, but can be deployed with moderate resources after initial development. While requiring machine learning expertise to develop, they are increasingly accessible through pre-trained models. Overall, these methods provide shallow control: system prompts alter generation via providing a specific context, while classifiers add a filtering layer over model outputs. Neither changes the model's underlying knowledge or tendencies. However, advances in classifier precision and accuracy simultaneously enhance the effectiveness of censorship mechanisms. Unlike parameter-modifying approaches, inference-time interventions can be deployed, modified, or removed without retraining, offering maximum flexibility for malicious actors to adapt to scrutiny or changing objectives.

**Real-World Evidence.** As illustrated in section 2.4, Elon Musk aligned the model *Grok* to reflect his personal political views. The resulting shifts in the model's behavior apparently stemmed from changes to its system prompt (Conger, 2025; Thompson et al., 2025), which, as previously discussed, serves as a fast and cost-effective alignment mechanism. However, some of these updates, including the instruction to be more "politically incorrect", also triggered antisemitic responses, the praise of Hitler, and the denial of Holocaust death tolls (Czopek, 2025; Field, 2025; Robins-Early & Gambino, 2025). This underscores which catastrophic side-effects such measures can have. Beyond prompt-based alignment, another immediate intervention can occur via real-time output filtering: Financial Times journalists observed how the Chinese model *Yi-large* suddenly changed a Xi-critical answer to an outright refusal response *after* generation – presumably, because the generated output triggered either key-word or model-based filtering methods put on top of the model environment (McMorrow & Hu, 2024). Similar observations have been reported for DeepSeek (Li, 2025).

## 4. Why It is Important to Talk About It *Now*

As alignment methods are further improved, they also provide increasingly powerful tools for censorship and manipulation at scale. Three parallel developments make this issue particularly pressing: AI systems are rapidly becoming important information sources for millions of users, the oligopolistic LLM industry creates concentrated points of control, and the global trend toward authoritarianism is extending into digital spaces.

### 4.1. Growing Use of AI Amplifies Misuse Impact

Conversational AI systems have achieved widespread adoption, with hundreds of millions of active users worldwide. Evidence suggests these systems are becoming primary information sources across diverse contexts: Tamkin et al. (2024) analyze usage data from Claude and find that users predominantly seek information for personal and professional purposes. Luettgau et al. (2025) document increasing reliance on AI chatbots for political information among UK users. A Reuters survey across six countries (Argentina, Denmark, France, Japan, the UK, and the US) reveals that weekly generative AI usage nearly doubled from 18% to 34% of respondents between 2024 and 2025, with information-seeking emerging as the dominant use case (Simon et al., 2025).

This shift toward AI-mediated information access is particularly concerning given concurrent improvements in model capabilities. Modern alignment techniques have produced systems that are highly eloquent and persuasive (Burtell & Woodside, 2023; Rogiers et al., 2024). As systems become more capable and more sophisticated in their outputs, they inspire greater user trust as indicated by increasing user numbers (Simon et al., 2025). The combination of widespread adoption as information sources and increasing user trust amplifies the potential impact of misuse: when millions rely on AI systems for information and decision-making, even subtle manipulations via alignment can have substantial societal consequences.

### 4.2. The LLM Oligopoly Creates Powerful Dependencies

The current LLM ecosystem is dominated by only a handful of companies, concentrated primarily in the United States and China (Vesteinsson et al., 2025). This oligopolistic structure, which is reinforced by high economic market entry barriers (Cottier et al., 2025), can yield systemic dependencies and power asymmetries (Winner, 1980), whereby only a small group of model providers and countries control the limited set of available foundation models and define the alignment choices embedded in them for everyone across the globe.

Additionally, individual state actors are able to compel global compliance with their rules through economic relevance. For example, if any model provider wants to access the (huge) Chinese market, it has to abide by its strict censorship laws. In the past, this has even led Google and other major tech companies to give in on censoring its products and search results (Dann & Haddow, 2008). To avoid the high cost and technical complexity of region-specific fine-tuning, model providers may be induced to directly apply the most restrictive regulations to *all* users in preemptive obedience. This allows the strongest and strictest govern-ments to dictate preferences for everyone (Sheehan, 2023).

### 4.3. Global Shift Toward More Authoritarian Regimes

Over the past decade, the world has witnessed a sustained democratic backsliding and global decline in civil liberties, with the average level of liberal democracy reverting to 1985 levels, as Nord et al. (2025) point out. Freedom of expression has deteriorated in nearly a quarter of all countries – setting a new negative record over the last 25 years – while restrictions on civil freedoms have worsened in dozens of nations, reflecting a broad erosion of the rule of law, as well as rising government censorship and repression. Disinformation and political polarization are accelerating these trends, with almost half of all autocratizing governments now actively spreading disinformation to undermine opposition and civil society, further entrenching authoritarian governance structures (Nord et al., 2025).

According to Freedom House (Vesteinsson et al., 2025), this especially affects digital liberties, with global internet freedom declining for 15 consecutive years – a trend including even half of the countries previously ranked as "free". Concretely, authorities increasingly influence online spaces to shape public discourse and promote state-sanctioned narratives in order to consolidate their power. In the age of AI, this of course also extends to LLM model providers, which face growing pressure by authoritarian regimes to "incorporate censorship of certain content, like criticism of the authorities" (Vesteinsson et al., 2025).

## 5. So, What Should We Do?

While completely eliminating the dual-use potential is impossible, several complementary approaches can reduce the likelihood and impact of deliberate misuse. We propose three key directions: establishing robust oversight mechanisms through transparency and verifiable alignment, maintaining pluralism and competition in the LLM ecosystem to prevent dangerous concentrations of power, and fostering awareness among both users and researchers. Lastly, we discuss why we do not call for stopping alignment research.

### 5.1. Oversight, Transparency, and Evaluation

The first way of mitigating deliberate misuse by model providers is public oversight and control. For democratic contexts, this is indeed sensible and has already been extensively discussed (e.g., Bogiatzis-Gibbons, 2024; Erman & Furendal, 2022; Seger et al., 2023). A prerequisite for public oversight, in turn, is transparency. Currently, however, it is very difficult to know which exact alignment practices were applied, as the underlying alignment policies, methods, datasets, and model internals of proprietary LLMs are often not publicly available (White et al., 2024). This in-

complete information prevents scrutiny and the conscious choice of models for users. Therefore, releasing this data – at least to independent auditors – would allow for systematic evaluation, comparison, and accountability regarding how alignment objectives are defined and enforced.[1] For example, the EU AI Act already mandates providers to disclose training methods and data to the authorities, which could be further extended accordingly.

However, even if respective requirements were implemented, the debate has yet to provide answers on how to address situations in which the model provider operates outside the effective reach of such legislation, or where a state itself becomes a malicious actor. Therefore, measures need to be developed to better evaluate even a black-box model's alignment objectively, without relying on the provider's cooperation. To this end, *standardized benchmarks* for information suppression and political bias would be valuable. Existing censorship benchmarks typically focus on narrow contexts, such as Chinese censorship (Ahmed & Knockel, 2024; Qiu et al., 2026) or information suppression about historical figures (Noels et al., 2025). However, comprehensive benchmarks covering diverse information types and regions are still needed. Similarly, while various benchmarks test political bias in LLMs using political orientation tests or sets of contentious questions (Bang et al., 2024; Peng et al., 2026; Rettenberger et al., 2025; Rozado, 2023), these typically examine bias only within specific national contexts, are focused on the right-left continuum, or rely on a limited selection of topics. Hence, we call for increasing efforts to develop and maintain standardized benchmarks that encompass *political contexts worldwide, account for authoritarian tendencies next to the right-left continuum, and remain dynamic* to reflect citizens' evolving realities. Developing such standardized and comprehensive benchmarks would be a step towards *verifiable alignment*, where users can independently verify exactly what values a model has been aligned with or which information gets suppressed.

### 5.2. Pluralism and Competition

Even the best and most complete information about a model's (mis)alignment does not help much if there is no alternative to choose from. Hence, the variety and plurality of models is a value in itself that needs to be upheld – not only to prevent a concentration of economical (market) power, but also of political and societal influence. After all, no single model can or will probably ever achieve total neutrality, objectivity, and fairness – not least, because there will never be a single truth or consensual understanding of what that should even look like in practice (Fisher et al., 2025). Therefore, just as in journalism, only the existence of diverse options will ultimately ensure to ap-

proximately achieve those objectives. Fisher et al. (2025) call this "Neutrality Through Diversity". We should thus prevent monopolies and one-sided dependencies from forming – be it from single model providers or countries – and embrace competition.

### 5.3. Awareness

Lastly, for users to actively consider potential censorship and manipulation when selecting or using LLMs, awareness is essential. Precedent for successful awareness-building exists in related domains. Studies on the effectiveness of media and digital literacy interventions to reduce the impact of misinformation show promise, with evidence indicating that even relatively short, scalable interventions can successfully improve users' ability to detect misinformation (Droog et al., 2024; Guess et al., 2020). Critical thinking has been identified as an essential skill for identifying manipulated or misleading information, with recommendations to integrate information literacy into educational curricula (Machete & Turpin, 2020; Romanishyn et al., 2025). These approaches offer a model for how interventions addressing LLM censorship and manipulation should be incorporated into broader digital literacy efforts, fostering individual empowerment to consciously choose models.

Just as we advocate for user literacy about AI risks, we alignment researchers must also demonstrate our own literacy: We must reflect on and communicate the potential risks of our work more genuinely, including in our publications. Many top-tier AI conferences like ICML now require "impact" or "ethics statements", asking researchers to consider the ethical and societal consequences of their research (Hecht et al., 2021). However, a general analysis of such statements for NeurIPS 2020 reveals that most authors engaged superficially, emphasizing positive over negative aspects of their work (Ashurst et al., 2022). Do et al. (2023) further conclude from interviews with researchers across multiple computer science sub-disciplines that while researchers acknowledge the need to consider unintended consequences, this reflection is rarely practiced. Among the reasons identified is the academic culture prioritizing rapid progress. In fact, the ICML author guidelines (including this year's) explicitly state:

> *"In many cases, where the ethical impacts and expected societal implications are those that are well established when advancing the field of Machine Learning, substantial discussion is not required, and a simple statement such as the following will suffice: [...]"* (ICML, 2026).

While subsequent instructions encourage consideration of particular risks, such statements lower the barrier for superficial engagement and discourage critical reflection. Only

---

[1]Such a 'certificate' could even be a competitive advantage.

through sincere and thorough engagement with the societal implications of our work can we, as alignment researchers, credibly advocate for the responsible development and deployment of AI systems.

### 5.4. Why We Do *Not* Call for Stopping AI Alignment Research

Given our warning about the dual-use potential of alignment methods, one could call for a complete halt to AI alignment research. However, we consider this an undesirable solution: while deliberate misuse of alignment methods is a real risk, this does not diminish the relevance and importance of alignment itself. There are already several reported cases of people driven to suicide due to a lack or faulty alignment of LLMs (Lovens, 2023; Rose, 2024; Stokel-Walker, 2025). Without appropriate safeguards, criminals and even terrorists can much more easily use these models to commit crimes and harm people (Chaudhry & Klein, 2025; Hendrycks et al., 2023). Moreover, when AI increasingly operates with greater autonomy and reduced human oversight, the stakes escalate considerably. Misalignment in such systems can produce immediate, direct consequences that may result in significant harm to humans, with severity proportional to the scope of deployment. This concern intensifies when considering the prospect of Artificial General Intelligence (AGI), which could execute numerous far-reaching tasks simultaneously and autonomously. In such a scenario, achieving the most secure and robust alignment possible transitions from important to absolutely critical (Bengio et al., 2025; Bostrom, 2014; Xu et al., 2025).

## 6. Alternative Views

Naturally, not everyone will share our conclusions. In the following, we therefore consider two divergent perspectives at both ends of the spectrum: first, the view that AI alignment should indeed be stopped; second, the argument that the risks we have outlined are exaggerated.

### 6.1. AI Alignment Should Be Stopped

Those who weigh the described dual-use risks more strongly might conclude that alignment research should stop altogether. The following argument could further support this position: From a libertarian perspective, one might ask to what extent it is even our responsibility to protect grown-up users from certain content at all costs. If someone explicitly requests specific content or wants to use it for a certain aim, it might not be on us to decide whether or not they should be able to obtain that information. Similarly, radical pamphlets or instructions can also be drafted in Word, but we do not require Microsoft to check every single document for harmful content. Consequently, despite well-intentioned objectives, current alignment methods may facilitate paternalistic sys-

tems that prioritize safety at the expense of freedom and self-determination (Bassini, 2025; Kalliris, 2024).

From this perspective, the existence of jailbreaks – targeted prompt manipulations that bypass model safety – can even be considered a form of 'freedom insurance'. Just as Virtual Private Networks (VPNs) enable individuals in repressive regimes to bypass internet censorship, functioning jailbreaks can guarantee access to information under restrictive systems. While these methods can undoubtedly be exploited for harmful or criminal activities – much like VPNs – their existence reflects a tension between security and freedom that alignment research should not resolve.

### 6.2. The Risk of Alignment Misuse is Exaggerated

On the other hand, others might consider our warning to be overstated.

**Alignment Techniques are Not the Only Means of Censorship.** If governments are keen on censoring AI, they do have other methods at hand next to alignment techniques. For instance, they could simply ban certain models from their territories, as some – particularly China, where Western LLMs are not available – have already done in the past (Davidson, 2023).

**Regulations Safeguard Against Misuse.** At least in liberal democracies, legal and even constitutional provisions, enforced by independent courts, prevent governments and usually also private actors to abuse or manipulate AI systems in ways that would harm basic rights, such as free speech or equality. A rising number of countries have implemented respective legislation on AI (Maslej et al., 2025), with the EU's Digital Services Act (DSA) and AI Act being the most prominent examples. They explicitly outlaw practices that could manipulate public opinion or discriminate certain groups, and implement mechanisms to mitigate such systemic risks, such as the reporting and transparency obligations stated before.

This, however, surely implies a strong rule of law, the willingness, as well as the technical and practical ability of states (Hadfield & Clark, 2026) to verify and enforce the compliance with those principles – even on foreign models. Notwithstanding the above, many countries have not yet implemented comparable legislation. In the U.S., the current administration explicitly rejected such proposals and repealed existing regulation (Robins-Early & Gambino, 2025). In fact, while constitutional provisions in the U.S. would likely – at least legally – prevent the government from implementing censoring mandates or compelled speech, private actors manipulating AI systems might even be protected to do so by the First Amendment to a certain extend (Bassini, 2025; Benson et al., 2025).

## 7. Conclusion

Alignment research is essential for safe AI systems, but we must acknowledge its dual-use nature. This paper has demonstrated how modern alignment methods – from pre-training filtering to inference-time controls – can be repurposed as tools for censorship and manipulation, with documented cases already emerging. The convergence of AI's role as increasingly important information source, market concentration among few providers, and global democratic backsliding creates unprecedented risks. What we build as safeguards today may become instruments of informational control tomorrow. Thus, we call on the community to engage genuinely with the dual-use potential beyond perfunctory ethics statements, develop standardized censorship and manipulation benchmarks for verifiable alignment, implement transparency and auditing mechanisms, and preserve competitive pluralism to prevent informational monopolies. Alignment remains essential, but must be pursued with clear recognition that it is a powerful tool that can serve both protection and oppression – depending on who wields it.

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
