# OpenReview forum: "Position: The Alignment Community is Unintentionally Building a Censor’s Toolkit"
_ICML.cc/2026/Position_Paper_Track — ICML 2026 Position Paper Track spotlight_

### Official Review · Reviewer_HTL2 · 2026-02-26

**Significance:** 4
**Argument Clarity:** 3
**Rating:** 5
**Confidence:** 3

**Questions:**

I would really like your opinion on my first point, and would greatly appreciate some changes in this re gard (although these might be rather deep changes).

**Alternative Views Section:**

Yes

**Compliance With Llm Reviewing Policy A Conservative:**

Affirmed.

**Discussion Potential:**

3

**Final Justification:**

I think the authors for carefully answering my questions and thank them for the promised changes, hoping they will adequately reflect our discussion. I thus raised my score accordingly.

**Paper Summary:**

This position paper highlights the fact that the alignment community builds a dual-use technology, that can be (and is) used for censorship and manipulation.

The beginning of the paper presents the alignment community as considering itself as "a safeguard to the risks of AI", with sometimes unintended harms of alignment methods. This leads to the main point of the paper: whatever the researchers intentions may be, alignment is a powerful tool (weapon?).

This dual-use lies in the fact that "AI alignment aims to make AI systems behave in line with human intentions and values", and so people can consider this technology dangerous if it is used by people that do not share their values. In particular, the two main actors (states and foundation model providers) have the power to "misuse" alignment tools. This "misuse" comes in two main forms: censorship and manipulation, and already takes place at varying degrees, for instance in China and the US (state actors), or for Elon Musk's Grok LLM (private actors).

Then, the inner workings of alignment are emphasized through a few examples:
- pre-training data filtering, that enables selective removal of unwanted content, or emphasis on desired one. This is document as being used for political prompts in Chinese LLMs.
- post-training preference alignment, achieved through (among other) RLHF, which requires less resources, and can steer models towards certain viewpoints (and again reported to being used in China).
- inference-time control, where the model itself is not modified but prompts/outputs are, e.g., after running through classifiers. This is reported as rather easy (and inexpensive) compared to other methods, and has been used in Grok.

The authors stress that now is an important time to highlight these issues, mainly because of the growing use of AI, the fact that LLMs are released by only a handful of very powerful actors, and that authoritarian regimes are on the rise worldwide.

This is followed by a call for action, built on the following principles:
- oversight, transparency and evaluation: entities should be as transparent as possible about their training methods and data, and benchmarks should be set up to evaluate what values a given LLM carries on given topics.
- pluralism and competition: there is no single truth or universal values, so diversity of models is required to reflect the diversity of values.
- awareness: LLMs should not be considered as conveying universal truth, and users should be aware of the potential biases (intentionally or unintentionally) embedded in them.

Finally, the papers present some alternative views, in particular:
- Alignment research should continue/increase pace because it is much needed
- Safeguards are already in place, and alignment is not required for censorship anyway.
- Alignment limits freedom (and jailbreaks might be welcome), so we should stop alignment research.

**Position:**

Yes

**Position In Title:**

Yes

**Related Work:**

3

**Strengths And Weaknesses:**

I believe this paper presents an important point. The dual nature of alignment research is clearly demonstrated and documented, and it is important that researchers in this field understand these stakes.

This being said, I have several concerns about the paper:

**1) The word "misuse".**
This word is used 29 times, and globally refers (in this paper) to alignment research being used by authoritarian regimes or companies that are not primarily interested in "public welfare".

I am not comfortable with this term, which tends to suggest that alignment research and LLMs more generally are tools that can be used for good or bad, thus disconnecting them from the social context they are developed in. It conveys the idea that Science is neutral, which is questioned by several Human and Social Sciences studies (e.g., Do artifacts have politics? Langdon Winner for an old reference).

I would rather not speak of "misuses" for systems so complex and deeply rooted in our societies such as LLMs. "misuse" suggests that there is a "right use" of LLM that is agreed on by the community, and I think such a statement requires evidence. Just because (some) researchers do not develop tools with some uses in mind does not mean these are misuses. Individual scientists claiming misuse of their work might have overlooked some aspects of the broader picture (which is why I think this paper is important, because the more this is discussed and the less researchers can claim misuse).

For instance, I am not sure one can claim that the atomic bomb was a misuse of civil nuclear energy research. Given how much research money actually comes from military budgets, I strongly prefer the wording "dual-use" (that you employ), which is upfront about the fact that there are both civil and military or "weaponized" applications (in a broad sense, i.e., used to assert dominance over other groups). Then, individual researchers can decide whether working on dual technologies aligns with their personal values. Alignment research is not "rightly used" if it supports an ideal universal benevolent value system and misused otherwise. It is fundamentally intended to promote a vision of society.

In a way, the paper acknowledges this social context, but does not go the full way as to recognize that this issue of power and control (economic, but not only) is at the core of LLM development. One may argue that LLM builders invest so much in them because they know this shaping power, and that LLMs (and alignment) are designed for it.

I acknowledge that this paper does recognize the complexity of the stakes to some extent (in particular the parts about the inexistence of an ideal universal value system, or the absence of the word "misuse" when refering to nuclear energy), but still somehow promotes a dichotomy between good and bad, which I believe weakens the argument.

I am a ML researcher myself, and I'm not making assumptions about the authors' background, but I feel relevant Science and Technology Studies related work and background is missing. I hardly know a few references in this field, yet I think discarding the word "misuse" and integrating a more structured critique about the neutrality of Science (building on Langdon Winner's work but also many others) would add depth to the discussion and strengthen the argument.

Similarly, I'm not so sure about the "unintentionally" in the title (but I'm not in this community).

**2) Alternative views are not all alternative.**
In the "alternative views" section, it's not always clear which is an alternative view, and what is the view defended by the authors (arguments against this alternative view). For instance, the "regulations safeguard against misuse" argument seems to directly integrate a critical paragraph about this view, whereas the other paragraphs do not seem to. Why be critical about the alternative view just in this case?

Similarly, it is not clear to me that all these are alternative views, as the paper does not call for stopping alignment research, so there's no tension between the fact that alignment is crucial to protect users from harm (it's even more or less what is stated in the conclusion). An alternative view may actually be to recognize LLMs as dangerous and hard to regulate, and stop developing them altogether (as could have been done for the nuclear bomb example taken throughout the article).

**3) Alignment is posed as necessary.**
The necessity of alignment is not so questioned ("its necessity is clear", or "alignment research is essential"), although there is a small part about this in the alternate view section. In the end, alignment necessity is hardly argued for. Beyond alignment, LLMs are presented as inevitable, and the possibility of just not promoting LLMs if they're not considered safe is never mentioned.

**4) Some arguments can be slightly manichean at times.**
This point is connected to my first point, but the arguments sometimes lack a bit of depth, and can be slightly manichean at times (good alignment: not telling how to make weapons, bad alignment: chinese censorship; security versus freedom...). Although there seems to be a conscious efforts to avoid this in many parts of the paper, some oversimplifying elements remain here and there.

**Support:**

3

---

> ### Author Rebuttal · Authors · 2026-03-30
>
> **W1**\
> Thank you for this valuable thought and also for pointing us to Winner’s paper, which is very relevant to this discussion! We agree that neither science nor technical applications can ever be fully neutral; they will always embody the biases of the social context in which they are set or developed. It is also true that alignment research was always meant to align AI systems with (certain) human values – which logically have to be defined somehow – and it thus never pretended to be (able of being) completely neutral in this regard. Accordingly, there is extensive literature on what values AI systems *should* be aligned to. However, and that is at the core of our concern, the community has never prominently considered that the underlying technical methods of how that is practically achieved are inherently value-agnostic, and thus *can* be (and, as we show, are being) used for various other purposes; including suppressing legitimate opinions, distorting facts, or unilaterally serving individual interests. These are (dual) use cases, which the research community developing these technologies has not yet actively discussed (as further outlined in our response to Reviewer 4MY6).
>
> Now, can we call these use cases “misuse”? That depends on the two semantic meanings of the word, according to the Cambridge Dictionary: “to use something (1) in an unsuitable way or (2) in a way that was not intended”.
>
> (1) Regarding the first meaning (unsuitability / malice): as outlined more extensively in response W3 to Reviewer cVtw, we agree that not every application of alignment diverging from Western liberal norms constitutes “misuse”. Thus, we admit that the word might appear too frequently in our text and sometimes in contexts where its grounding is not made sufficiently explicit. We therefore commit to substantially reducing its use in the revision and instead giving preference to the more descriptive term "dual-use", as you suggested, and reserving stronger normative language for cases where it is properly anchored. However, we maintain that *some* applications are indeed objectively malicious in a meaningful and defensible sense: specifically those that violate internationally recognized universal human rights and exploiting power asymmetries, which we commit to define more thoroughly in Section 2.2.  Regarding the question of power and control, we found Winner particularly interesting and will include him for that in the revision.
>
> (2) Now, concerning the second meaning of the word “misuse” (intention): The question is whether alignment researchers are already *purposely* designing tools for the use cases we are warning about (like R. Moses with his low bridges mentioned in Winner’s article). We are confident in negating that for the prevailing opinion within the alignment research community: There is no paper defending or even calling for using alignment in the described ways. Instead, there is still a prevailing (and largely unquestioned) self-assessment of being benevolent when it comes to alignment techniques. This is exactly what the term "unintentionally" in our title reflects: a community-level failure to question the political potency of our work. You are right that researchers from our scientific community have probably largely invested in alignment research because they do understand (the need for) its shaping power – however, with other (even orthogonal) aims in mind than what they might have ended up facilitating. Hence, as long as this is the case, from the prevailing point of view of our community, yes, applying our technologies for censorship and manipulation is indeed more an unintended “misuse” than only an (explicitly or implicitly accepted) “dual-use”. Thus, we want those who build the technologies to be aware of what they are capable of.
>
> Hence, for those specific cases, we tend to retain the word "misuse" in the relevant sections – consciously accepting that this is a non-neutral position, which is however rooted in the prevailing urge to build „helpful, honest, and harmless“ AI systems (Askell 2021) within the community we address.
>
> We found this discussion very stimulating and will add the core arguments and related literature in the revision
>
> **W2**\
> On only defending against one alternative view:
> True. We will comment on all.
>
> On stopping LLM development altogether:
> Surely, this would be the most consequential solution to the risks posed not only by alignment, but by LLMs altogether. However, we would question whether this is a reasonable call given the current AI race and initiatives that already failed to hold or pause AI development. In short, it is probably already too late to do so. But if you feel strongly about this, we would be open to adding this to the alternative views.
>
> **W3**\
> You are right. We will move the necessity claim from the alternative views to the main body (see also our response to W2 by Reviewer cVtw)
>
> **W4**\
> See our response to W1 here and to W3 with Reviewer cVtw

---

> > ### Author Rebuttal · Reviewer_HTL2 · 2026-04-01
> >
> > Thank you for your careful reading and taking my comments into account, especially the one on misuse and "manichean" aspects.
> >
> > I agree with you that from the researcher's (or say, most individual) point of view you could consider it misuse, but from the point of view of the companies that fund and develop these technologies it is less clear to me that there is no intent to exploit power asymmetries for instance. To me, part of the "community-level failure" also rests in failing to recognize larger-scale incentives that may contradict one's values. Yet, I don't want to argue against accepting this paper on this (as long as non-neutrality of the science we do is acknowledged, like it seems you intend to do from the rebuttal).
> >
> > In the end it depends on which scale you look at things, and I do believe that raising overall awareness about these issues is a good thing, and the paper does a good job at it.
> >
> > W2: I believe I'd prefer to have this expicitly written down as an option indeed, not necessarily as a call or a desirable thing, but as a valid alternative view, which poses other problems as you mention (e.g., hard to do without massive coordination). Yet, even after reading this paper, it's not clear to me how alignment research can avoid dual-use, so it's the same problem in the end: people that want to weaponize it are free to do so, which is not a very satisfactory option either. I don't feel so strongly about it to explicitly ask for it though. If you find a way to integrate it that suits you then I'd be glad to see it in the final paper, but if you're reluctant to do so feel free not to incorporate it.

---

### Official Review · Reviewer_cVtw · 2026-03-06

**Significance:** 3
**Argument Clarity:** 3
**Rating:** 5
**Confidence:** 5

**Questions:**

N/A

**Alternative Views Section:**

Yes

**Compliance With Llm Reviewing Policy A Conservative:**

Affirmed.

**Discussion Potential:**

4

**Final Justification:**

The authors addressed my concerns.

**Paper Summary:**

This paper makes obvious the dual-use risks of AI alignment technology. Their position is that (1) we should raise awarness on the power structures behind so-called aligment efforts to make it clear that the can and are being used for means that the original alignment community did not intend (e.g., state actors and censorship) (2) in response to this we can push for more transparency on how foundation models are being shaped and encourage pluralism where values conflict (among other suggestions).

**Position:**

Yes

**Position In Title:**

Yes

**Related Work:**

3

**Strengths And Weaknesses:**

## Strengths

Although I have some reservations about how the argument was conveyed and whether this position is novel enough to warrent conference inclusion (detailed below), I generally believe that (A) this is an important topic to discuss at the conference (B) even if this topic is already treated in other places, many papers on the topic are important for raising awareness (C) the authors opinions, if read suffeciently critically, could create a lot of discussion.

Beyond the above, I think the authors did a good job at what many safety and security papers which mention dual-use risk fail to do: provide concrete examples and threat actors. In particular I can see this paper as being a resource for researchers wanting to reach for credible threat models (state actor censorship or LLM provider ideological hegemony). For that reason alone I recommend this paper for acceptance as I don't think the community has too many examples they can circulate to motivate their work.

## Weaknesses

### Novelty

I believe from the beginning the value alignment community has been aware of and struggled with the problems raised in this paper. On the one hand, the value alignment community as represented by, say, lesswrong and AI alignment form is a different community than ICML and so perhaps main mainstream PhD students working on ``alignment'' techniques are unaware of these arguments. On the other, we should be aware that these techniques come from those communities (e.g., Christiano 2017; Askell 2021 are both representative here) where these same problems of value alignment and dual use risk have been discussed albeit in a quasi-academic setting.

Perhaps the authors believe they have addressed this to scoping the paper's frame to be that of a conversation emerging in direct response to this happening in non-hypothetical settings today. But I still think that there is perhaps a missing effort to investigate sources from the value-alignment community like [1], [2], and [3] which I will discuss in my next point. Either way, the authors should take a careful look at the text to see if there are any places that they might accidently be suggesting that serious alignment researchers don't _already_ consider dual use risk (This is why Yudkowsky left the research community after all (reminiscent of Grothendiek quite a theme!)).

[1] Hadfield-Menell, Dylan, Anca D. Dragan, Pieter Abbeel, and Stuart Russell. "The off-switch game." In AAAI Workshops. 2017.
[2] Hadfield-Menell, D., Russell, S. J., Abbeel, P., & Dragan, A. (2016). Cooperative inverse reinforcement learning. Advances in neural information processing systems, 29.
[3] Corrigibility. N Soares, B Fallenstein, S Armstrong, E Yudkowsky. AAAI Workshop: AI and Ethics, 2015

### Missing alternative view: Core human values

Given my coloring above, I think 6.1 is a strawman / false dichotomy. See the classic picture of misalignment with human values that may emerge naturally in [4]. For example lets take resource aquisition and an instrumental goal of a misaligned AI - I don't think that anyone is arguing that preventing catastrophy from unchecked resource aquisition should be done despite the risks of misuse. Instead I think the more natural alternative position is that the authors are discussing **alignment with conflicting human values** which is not as important as **alignment with general values we all share** such as wanting to survive on this earth which instrumental resource aquisition is at odds with.

[4] Omohundro, S. M. (2018). The basic AI drives. In Artificial intelligence safety and security (pp. 47-55). Chapman and Hall/CRC.

### Conflating Misalignment, Misuse, and Value Conflicts

I find the authors language very value-laden with a particular ideological position that prevents them from seeing that alignment is neutral w.r.t ideology. We should expect alignment technolgy to be used to align AI's with a variety of values that conflict. They give an example of this when they paint the picture of foundational model training in China. In China, they are very much concerned with value alignment for social cohesion and stabiltiy - free speech is not a shared value with the west and so naturally value alignment for China conflicts with value alignment of foundation models in other places. I am very reluctant that we call conflicting alignment settings misuse.... perhaps the authors can make more room for alternative ideological positions and conflicting value alignments as seperate from misuse? In sum, I really don't think that censorship is misuse or dual use risk when there are lots of reasons why people would want censorship, for example in the country I am from we have censorship laws preventing hate speech which I find is aligned with the broadly western enlightment sensability I subscribe to.

Now the authors are right to point out and raise awarness of the power structures behind what values are used for alignment and for what purpose: value alignment can be for oppression and the violation of human rights. I just don't know that value alignment belongs to the same category of things as safety, morality, or ethics.

It is hard to express this criticism without asserting my opinion so it only factors slightly into my score for giving a lower support and argument clarity score due primarily to lack of nuance on the part of the authors.

**Support:**

3

---

> ### Author Rebuttal · Authors · 2026-03-30
>
> Thanks for this valuable feedback!
>
> **W1**\
> On the general question of novelty, we would like to refer you to our detailed response to Reviewer N2pD. Here we focus on what is particular to your critique. We thank you for pointing us to the cited papers. They are actually a great help to contrast our contribution!
>
> [2] proposes cooperative inverse reinforcement learning (CIRL) to address the threat that humans may mis-state their objectives, leading to unintended AI misbehavior. Our threat model is orthogonal: we are not concerned with a human who unintentionally mis-states their objectives, but with one who deliberately teaches the model malicious behavior. The very mechanics that make CIRL powerful also become powerful tools for a state actor systematically instilling censorship objectives. That this dual-use potential goes undiscussed in this paper is precisely the gap our work addresses and wants to draw attention to.
>
> [1] and [3] are concerned with unintended misalignment: an AI that resists human correction or shutdown because rational goal-directed behavior structurally produces self-preservation incentives. The adversary in their threat model is the *AI itself*. Our threat model is the inverse: improving AI alignment methods makes the AI work more as intended, which is precisely the danger when those operators have malicious goals.
>
> We do not want to claim to be the first to raise *AI misuse concerns in general*, but we do fill a concrete gap: systematically showing how specific *technical alignment methods* enable this misuse, with documented evidence that it is already happening, embedded in an analysis of the social, economic and political developments that make this discussion urgent. We hope you agree that this is a distinct and valuable contribution. We will make this distinction to general AI risks and the value-alignment community discussions more explicit.
>
> **W2**\
> We agree that there are some general values, we or the research community can all consent on; this includes preventing catastrophe by unchecked (agentic) AI pursuing its own interests, as you mention, but also universal human rights (as we argue in the next paragraph). This is what Section 6.1 seeks to stress: the stakes of halting alignment research would be too high. The problem is that there is also no “middle way”: it is not possible to stop alignment *only* for conflicting human values, because regardless of its aim, alignment is always based on the same underlying technical methods. Basically, it is “all or nothing”.
>
> However, we do concede that Section 6.1 can be considered a strawman, since we explicitly do not call for halting alignment, although we do say that it can’t go on as it is. Hence, also regarding W3 of Reviewer HTL2, we propose to incorporate it into the paper’s main body instead in order to substantiate our claim in the introduction that the necessity of alignment is “clear”.
>
> **W3**\
> Thank you for this valuable comment! We fully agree that not every application of alignment that conflicts with Western liberal norms constitutes “misuse”: different societies hold legitimately different values (as illustrated by your own example of differing approaches to free speech and hate speech, on which even Western countries like the U.S. and Europe do not fully agree), and the existence of such disagreement is precisely why we advocate for model pluralism in Section 5.2.
>
> However, we do maintain that some applications of these techniques are not merely value-divergent but malicious misuse in a meaningful and defensible sense: Specifically, if they (i) violate internationally recognized human rights norms (which you also mention), including freedoms of expression, thought, and access to information as enshrined in the Universal Declaration of Human Rights, to which virtually all states are formally committed, or (ii) serve the private interests of a small number of powerful actors at the expense of the billions of users who have no visibility into, or recourse against, those decisions (which is a more systemic rather than value-based issue). These are the concrete threats we want to raise awareness for, and we intend the terms “censorship” and “manipulation” to be understood in this narrowly defined way. However, we acknowledge that the paper has not yet made this sufficiently explicit.
>
> Hence, we do agree with your point that we should be more nuanced in this regard. Therefore, we will expand Section 2.2 accordingly by defining the normative claims this paper does and does not make, including (1) a disclaimer about legitimate value divergences, and (2) more explicitly describing the concrete scope of what we mean by “malicious” or “misuse”, as outlined above – and preferably using the more neutral term “dual-use” in other cases (as also concluded in our response to weakness 1 by Reviewer HTL2).
>
> If you accept these improvements, we hope you consider raising your argument clarity and support scores accordingly.

---

> > ### Author Rebuttal · Reviewer_cVtw · 2026-04-01
> >
> > I appreciate the thorough respone here and to the other reviewers.
> >
> > I am going to raise my soundness and clarity scores and remain confident in my other scores (recommendation for acceptance). In particular, I appreciate your position with much more clarity now when contrasted with other works on alignment I am familiar with. Your response reveals very clearly how the paper is positioned.
> >
> > I appreciate the additional commitement to improving the papers quality in both this rebuttal and the other authors rebuttals.
> >
> > I have also raised my discussion potential scores as I am sure many people will have strong opinions as the reviewers did so I feel as the the ``position'' part of the paper was well done.

---

### Official Review · Reviewer_4MY6 · 2026-03-13

**Significance:** 2
**Argument Clarity:** 3
**Rating:** 5
**Confidence:** 4

**Questions:**

My primary concern regarding this position paper is that the stated position seems obvious and I perceive "alignment techniques can also be used to align models to unsafe/malicious behavior" as a well-known information within research community. Despite the fancy phrasing "The Alignment Community is Unintentionally Building a Censor’s Toolkit", I hardly find this position to inspire useful discussion.

But my perception might not be precise always. If the authors have any evidence/indicator suggesting this being unknown, I would like to hear it. I will also try to incorporate the other reviewers opinion regarding this matter.

**Alternative Views Section:**

Yes

**Compliance With Llm Reviewing Policy A Conservative:**

Affirmed.

**Discussion Potential:**

3

**Final Justification:**

For me personally, the main position of this submission "alignment techniques can also be used to align models with unsafe/malicious behavior" has long been assumed as sth that pretty much everyone (in the relevant fields) knows. This factor aside, this submission actually did a fairly good job in the discussion and analysis.

Given the perception of the other reviewers (and also thanks to the authors for help highlighting/summarizing angles they discussed that other reviewers find especially compelling), I believe it is safe to say that my original view was not a good representative of the fields and this would inspire probably a fair portion of relevant researchers. Thus I am raising my overall assessment to 5. accept and Discussion Potential to 3. good.

**Paper Summary:**

The submission advocates that many alignment technology can also be repurposed for censorship and manipulation.

It discussed ways that alignment techniques can be used for censorship/manipulation and also discussed indicators of existing usage.

**Position:**

Yes

**Position In Title:**

Yes

**Related Work:**

2

**Strengths And Weaknesses:**

The stated position is well supported through reasoning and the presentation is clear.

My primary concern regarding this position paper is that the stated position seems obvious and I perceive "alignment techniques can also be used to align models to unsafe/malicious behavior" as a well-known information within research community. Despite the fancy phrasing "The Alignment Community is Unintentionally Building a Censor’s Toolkit", I hardly find this position to inspire useful discussion.

**Support:**

3

---

> ### Author Rebuttal · Authors · 2026-03-30
>
> Thanks for your important engagement with our work! We take the novelty concern seriously and would like to address it directly, as we believe there is an important distinction between what prior work has discussed and what we contribute.
>
> Our unique contributions are as follows:
> 1. We are among the first to explicitly establish that *technical alignment methods* pose a dual-use risk and are hence the *means to realize* AI misuse by malicious actors. Previous research talks about the dual-use risk of *AI systems in general* – not the technical alignment methods – from a high-level and often from a hypothetical angle or discusses how (more agentic) AI itself poses a risk as it could (accidentally) pursue goals that are misaligned with human values (see citations provided by Reviewer cVtw). Especially the potential occurrence of this value-misaligned AI is a threat model that is orthogonal to our threat model which concentrates on single actors and how they could use their extraordinary power to repurpose technical AI alignment methods for their own goals. Hence, to the best of our knowledge, there is no systematic academic work that states the dual-use risk of the *technical alignment methods themselves*. While some papers do examine isolated aspects of censorship in specific models or regions (as we discuss in Section 1 and 5.1 second paragraph), none systematically frames these as instances of a broader dual-use risk inherent to alignment methods themselves.
>
> 2. We develop a comprehensive framework that specifically maps the alignment methods – split along their phase (pre-training, post-training, inference time) – to their dual-use potential as rated by the factors of access requirements, computational resources, technical expertise, as well as ease and depth of modification.
> 3. We directly connect real-world evidence to the dual-use potential of these single alignment methods (which we have not seen aggregated in this way before and what Reviewer cVtw specifically values)
> 4. We are the first to characterize relevant actors (states and model providers) with respect to how easy they could use their power to repurpose alignment techniques to their will, potentially at the disadvantage to the public. As Reviewer cVtw points out: “this paper [is] a resource for researchers wanting to reach for credible threat models (state actor censorship or LLM provider ideological hegemony). For that reason alone I recommend this paper for acceptance”
> 5. We propose concrete (and novel) counter measures that e.g. Reviewer N2pD finds compelling
> 6. We connect the dual-use potential of technical alignment methods to the current social, economical and political developments that make this discussion very urgent. As Reviewer N2pD points out: The paper “also hammers the point that the discussion on the topic is needed now to restrict further damage through state-sponsored censorship and manipulation”
>
> Also, on the question of our position’s obviousness, significance and discussion potential, we naturally disagree: We strongly believe that the point we raise is a pressing problem that – in this form – has received little to no attention in the technical alignment community so far. Although one might perceive it as “obvious”, the general issue of the dual-use potential of *alignment techniques* has not really been discussed in our field so far, which is also reflected in the lack of relevant literature on this topic. This practical experience inspired us to write this paper, in order to raise awareness for the topic.
>
> To support our position, we also want to refer to the other reviewers who awarded consistently high scores on both dimensions (significance: 4, 4, 3; discussion potential: 4, 3, 3), with Reviewer cVtw explicitly recommending acceptance on these grounds: “I generally believe that (A) this is an important topic to discuss at the conference” and that our opinion “could create a lot of discussion”. Similarly, Reviewer HTL2 says: “I believe this paper presents an important point. The dual nature of alignment research is clearly demonstrated and documented, and it is important that researchers in this field understand these stakes.”
>
> We hope this response helps align our views, and we genuinely appreciate your willingness to revisit your assessment. Thanks a lot!

---

> > ### Author Rebuttal · Reviewer_4MY6 · 2026-04-02
> >
> > For me personally, the main position of this submission "alignment techniques can also be used to align models with unsafe/malicious behavior" has long been assumed as sth that pretty much everyone (in the relevant fields) knows. This factor aside, this submission actually did a fairly good job in the discussion and analysis.
> >
> > Given the perception of the other reviewers (and also thanks to the authors for help highlighting/summarizing angles they discussed that other reviewers find especially compelling), I believe it is safe to say that my original view was not a good representative of the fields and this would inspire probably a fair portion of relevant researchers. Thus I am raising my overall assessment to 5. accept and Discussion Potential to 3. good.

---

### Official Review · Reviewer_N2pD · 2026-03-13

**Significance:** 4
**Argument Clarity:** 4
**Rating:** 5
**Confidence:** 4

**Questions:**

-

**Alternative Views Section:**

Yes

**Compliance With Llm Reviewing Policy A Conservative:**

Affirmed.

**Discussion Potential:**

4

**Final Justification:**

I will keep my original score. I started with a positive assessment (based on the strengths), and it has not changed after the rebuttal.

**Paper Summary:**

The authors argue that alignment methods in the wrong hands can be easily used for censorship and manipulation. They argue that alignment values might be decided by potentially bad-faith actors, which they identify as state and foundation model providers. They also provide real-world examples where censorship and manipulation through AI models are already in play. They identify three methods to misuse alignment techniques, namely, pre-training data filtering, post-training preference alignment, and inference-time control, while providing evidence of existing misuse. Next, they study the impact of such misuse at a global level and provide recommendations to restrict the misuse of alignment methods.

**Position:**

Yes

**Position In Title:**

Yes

**Related Work:**

4

**Strengths And Weaknesses:**

**Strengths**
1. The paper presents its ideas in a clean and concise manner. The authors state their position clearly and present ample evidence to argue their case.
2. The alternative views have been discussed adequately, and the authors make good recommendations to curb the misuse of AI alignment.
3. The use of existing real-world examples, especially in Sections 3 and 4, emphasizes the significance and relevance of the problem. It also hammers the point that the discussion on the topic is needed now to restrict further damage through state-sponsored censorship and manipulation.

**Weaknesses**
1. It should be noted that the misuse of AI alignment is a complex problem, especially due to the involvement of the state as a potential adversary. While the plan of action seems reasonable, it depends heavily on the cooperation and goodwill from both the state and the foundational model providers. This dependence puts a question mark on the implementation and efficacy of the suggested plan of action.

**Support:**

4

---

> ### Author Rebuttal · Authors · 2026-03-30
>
> Thanks for your engagement with our paper and for acknowledging the clarity of our arguments, the value of our proposed counter measures and the provided evidence!
>
> Regarding the point of cooperation goodwill you raised: We totally agree that tackling the dual-use potential of alignment techniques can be challenging given that state-actors might be involved. This is precisely why in the second paragraph of Section 5.1 we discuss what should be done if either the state itself becomes a “malicious actor” or model providers operate outside the effective reach of the proposed AI legislation. We suggest developing objective, standardized benchmarks that would be a step towards *verifiable alignment*, where users can independently verify exactly what values a model has been aligned with or which information gets suppressed. We further call for more model pluralism and market competition (Section 5.2) to allow users to choose more freely, reducing the concentration of power from state actors or model providers.
> Of course, it remains true that if a state actor strongly seeks to suppress access to certain models or access to those benchmark information, it has means at hand to do so (although this would also cut them off from access to frontier AI systems). However, even then, people might still find ways to circumvent state oppression, as they do via VPN for accessing other restricted websites – making our suggestions above still valuable.
> Lastly, in Section 5.3 we also call for awareness through improved user literacy about the topic and we also ask the research community to more thoroughly discuss the dual-use potential of alignment improvements in their impact statements, which is somewhat independent of the goodwill of state actors and the model developers.
>
> As you say, the complexity of the topic shows how important it is to discuss the topic now, which is what we would love to do at ICML 2026.

---

> > ### Author Rebuttal · Reviewer_N2pD · 2026-04-02
> >
> > Thank you for your response. I have gone through the other reviews as well and have decided to keep my original positive score.

---

### Decision · Program_Chairs · 2026-04-30

**Decision:**

Accept (spotlight)

**Comment:**

Overall, this is an excellent position paper that many in the research community need to read.
The key position here on the duality of alignment training is non obvious, and will likely surprise many.
There has been extensive discussion between the authors and the reviewers (really excellent to see), with the result that all reviewers have finalized on the "accept" decision.

I personally have learned quite a bit from reading this paper, and will be actively promoting it to others in my research community as a viewpoint that we should all be aware of.